# Thermoanalytical and X-ray Diffraction Studies on the Phase Transition of the Calcium-Substituted La_2_Mo_2_O_9_ System

**DOI:** 10.3390/ma16020813

**Published:** 2023-01-13

**Authors:** Artūras Žalga, Giedrė Gaidamavičienė

**Affiliations:** Department of Applied Chemistry, Faculty of Chemistry and Geosciences, Vilnius University, Naugarduko Str. 24, 03225 Vilnius, Lithuania

**Keywords:** sol-gel synthesis, phase transition, thermal analysis, X-ray diffraction, Rietveld refinement

## Abstract

An aqueous sol-gel preparation technique was applied for the synthesis of calcium-substituted lanthanum molybdate with the initial composition of La_2–x_Ca_x_Mo_2_O_9–x/2_. The influence of the substitution effect, which plays a crucial role in the formation of final ceramics, was investigated. The thermal behavior tendencies of phase transition at elevated temperatures from the monoclinic crystal phase to cubic as well as reversible transformation were identified and discussed in detail. It was proved that the phase transformation in the obtained mixture significantly depends only on the impurities’ amount, while the partial substitution by calcium atoms above the value of x = 0.05 does not create a homogeneous multicomponent system for La_2–x_Ca_x_Mo_2_O_9–x/2_ composition.

## 1. Introduction

Since the discovery of enhanced ionic conductivity for the La_2_Mo_2_O_9_ compound by Lacorre in 2000 [1], the efforts of application [2] for this system in different electrochemical devices have continuously increased [3]. Oxygen pumps, sensors, and solid oxide fuel cells (SOFCs) [4,5,6,7] are only a few types of equipment where lanthanum molybdenum oxide can be successfully applied. Despite a reversible phase transformation [8,9] above 540 °C from a low-temperature form α-La_2_Mo_2_O_9_ [10] to a high-temperature form β-La_2_Mo_2_O_9_ [11], its chemical stability [12] under air atmosphere in the range of temperature from 600 °C to 1000 °C creates the conditions for using this compound as a solid electrolyte of oxygen ions [13]. Moreover, the densification [14] of the corresponding ceramic could be successfully applied below the temperature of 1200 °C while creating desirable surface and crystalline properties [15,16]. The synthesis technique [17,18,19] that allows the preparation of the initial mixture of lanthanum and molybdenum oxides also plays an important role during the formation of the final ceramic at high temperatures. However, the molar ratio of initial metals remains the main factor that determines the formation of the La_2_Mo_2_O_9_ composition. This is the reason why the partial substitution [20,21,22] of either lanthanum [23,24,25] or molybdenum [26,27,28,29] leads to the crystallization of side phases [30,31], which significantly affects the physical properties [32,33] of the corresponding compound. This effect is directly related to both the amount of the La_2_Mo_2_O_9_ phase in the final ceramic mixture and the increased stabilization of the cubic phase at room temperature. Therefore, the main aim of this work was to study the dependence of the phase transition of La_2_Mo_2_O_9_ ceramics on the degree of calcium substitution in the corresponding system.

## 2. Materials and Methods

La–Ca–Mo–O tartrate gel precursor for La_2–x_Ca_x_Mo_2_O_9–x/2_ ceramic was prepared by an aqueous sol-gel synthesis using tartaric acid as a chelating agent that interacts as a ligand at the molecular level with the reaction mixture during both the dissolution in water and either sol or gel formation. The general scheme of this experiment is illustrated and presented in Figure 1.

Lanthanum (III) oxide (La_2_O_3_, 99.99% Alfa Aesar), molybdenum (VI) oxide (MoO_3_, 99.95% Alfa Aesar), and calcium (II) nitrate tetrahydrate (Ca(NO_3_)_2_·4H_2_O 99.98% Alfa Aesar) were used as starting materials and weighed before the dissolution procedure according to the desired stoichiometric ratio. It should be noted that, despite the high purity of the lanthanum (III) oxide, it was additionally heat-treated at 1000 °C for 5 h because of its tendency of the reaction with humidity and carbon dioxide from the air. In this case, even a slight deviation in the lanthanum amount from the ideal composition for La_2_Mo_2_O_9_ ceramic creates conditions for the formation of impurity phases such as La_2_Mo_3_O_12_ or La_2_MoO_6_ [34]. Nitric acid (HNO_3_ 66% Reachem (Mississauga, Canada)), distilled water, and concentrated ammonia solution (NH_3_ · H_2_O 25% Penta (Prague, Czech Republic)) were used as solvents and reagents to regulate the pH of the solution. Tartaric acid (L–(+)–Tartaric acid (C_4_H_6_O_6_) (TA) ≥ 99.5% Sigma-Aldrich (Darmstadt, Germany)) was applied for escalation of solubility via coordination of starting compounds in the reaction mixture, especially during the pH changes and evaporation before sol-gel formation. The mechanism of the corresponding chemical process in the frame of the aqueous tartaric acid-assisted synthesis for the preparation of the La–Mo–O gel precursor was discussed in our previous work [35]. Finally, the obtained La−Ca−Mo−O tartrate gel precursor for La_2−x_Ca_x_Mo_2_O_9−x/2_ ceramics was heat-treated for 5 h at 1000 °C in the air atmosphere.

The thermal analysis of heat-treated powders was performed with TG–DSC, with a STA 6000 PerkinElmer instrument using a sample mass of about 20 mg and a heating rate of 40 °C min^–1^ under an airstream of 20 cm^3^·min^–1^ at ambient pressure. The heating and cooling cycle was fulfilled twice from 300 °C to 800 °C and from 800 °C to 300 °C. The sample mass, heating rate, atmosphere, and its flow rate were selected empirically during numerous tests to ensure the best signal peak efficiency and to minimize the noises and background signals, which occur because of the influence of the corundum crucible and equipment limits. The characteristics of the phase transition peak were evaluated in the ranges of temperature from 530 °C to 600 °C for heating and from 560 °C to 490 °C for the cooling regime. X-ray diffraction (XRD) patterns were recorded in air at room temperature by employing a powder X-ray diffractometer Rigaku MiniFlex II using CuK*α*_1_ radiation. XRD patterns were recorded at the standard rate of 1.5 2*θ* min^–1^. The sample was spread on the glass holder to obtain the maximum intensity of the characteristic peaks in the XRD diffractograms. The Rietveld refinements of the obtained XRD patterns were performed using X’Pert HighScore Plus version 2.0a software.

## 3. Results and Discussion

### 3.1. Thermal Analysis

In this work, thermal analysis as a powerful investigation technique was used for a detailed investigation of the crystal phase transition from the monoclinic α-phase to cubic β-phase and from the cubic β-phase to monoclinic α-phase in the La_2–x_Ca_x_Mo_2_O_9–x/2_ ceramic system. An example of a differential scanning calorimetry (DSC) curve for the La_1.95_Ca_0.05_Mo_2_O_8.975_ compound is presented in Figure 2. The corresponding results for other samples are presented in the Appendix A. Meanwhile, the data of the phase transition during the repeated heat treatments are collected in Table 1.

It is seen from Table 1 that the enthalpy values of the first heating cycle are slightly lower, especially in the cases with a smaller amount of calcium ions, compared with the second one. The reversible stabilization of the cubic phase at room temperature after partial transformation from the monoclinic α-phase determines the main reason for such behavior. According to the measurement conditions, the second heating cycle corresponds to phase transition energy more precisely. Therefore, the representation of the tendency of enthalpy change of only the second heating and cooling cycles according to the substitution degree of calcium ions is shown in Figure 3 and Figure 4. The decrease in the tendency of phase transition enthalpy by increasing the calcium amount in the corresponding system is directly related to the amount of the monoclinic crystal phase of the La_2_Mo_2_O_9_ compound. Nevertheless, during the cooling stage, the increased enthalpy of the phase transition in the La_1.9_Ca_0.1_Mo_2_O_8.95_ sample shows that the reduction of the La_2_Mo_2_O_9_ phase is not the only factor that determines the energetics of the phase transition.

This phenomenon could be explained either by the increase in the amount of the monoclinic phase or by the influence of calcium ions on the formation of side phases in the final ceramic mixture. By further increasing the concentration of calcium ions in the La_2–x_Ca_x_Mo_2_O_9–x/2_ system, the enthalpy of the phase transition starts to decrease, and this result is directly related to the decrease in the amount of the crystalline phase for La_2_Mo_2_O_9_ in the final ceramic.

Summarizing the phase transition results obtained from cooling cycles, it can be concluded that homogeneous substitution by Ca^2+^ ions in the La_2–x_Ca_x_Mo_2_O_9–x/2_ system takes place up to the value of x = 0.05. In this case, the phase transition mainly depends only on the amount of the monoclinic crystal phase in the La_2_Mo_2_O_9_ ceramic homogeneously substituted by Ca^2+^ ions. The increase in enthalpy values of the phase transition for La_2–x_Ca_x_Mo_2_O_9–x/2_ (x = 0.10 and 0.15) samples during the cooling stages could be explained by the side phase effect, which increases the amount of pure La_2_Mo_2_O_9_ compound and its monoclinic phase in the final ceramic mixture.

### 3.2. X-ray Diffraction

In order to prove the crystalline composition in the obtained La_2–x_Ca_x_Mo_2_O_9–x/2_ system, the XRD analysis of the corresponding ceramic was also performed. The XRD patterns of all samples that correspond to the data collected in Table 2 are presented in the Appendix B.

Meanwhile, Figure 5 is consistent with XRD data, which show the formation process and trends of La_1–x_Ca_x_Mo_2_O_9–x/2_ and CaMoO_4_ crystalline phases. As it seen, the enthalpy of the phase transition for La_2_Mo_2_O_9_ mostly depends on the amount of the monoclinic phase in the ceramic mixture. This assumption is confirmed by the increased stabilization of the cubic phase up to 48.0% even after insignificant substitution of lanthanum by calcium ions in the La_1.999_Ca_0.001_Mo_2_O_8.9995_ system.

Nevertheless, by a further increase in the substitution degree of lanthanum by calcium (x = 0.01 and 0.05), the amount of the monoclinic phase for the La_2_Mo_2_O_9_ compound slightly increases; however, the trend of phase transition enthalpy change remains in a decreasing manner as concluded from Figure 3. Considering the fact that the amount of impurity phases in the obtained ceramics is really small, this decrease in the enthalpy of phase transition is basically determined by the increase in the concentration of the mixed-phase La_2–x_Ca_x_Mo_2_O_9–x/2_. This statement is partially confirmed by the XRD diffractogram of the Ca_1.9_Ca_0.1_Mo_2_O_8.95_ compound, in which quite a significant amount of the crystalline side phase for the CaMoO_4_ was identified. It seems that this impurity phase effect reduces the amount of the La_2–x_Ca_x_Mo_2_O_9–x/2_ homogeneous phase in the mixture and creates conditions for the formation of pure La_2_Mo_2_O_9_ compound. This explains the increase in the phase transition enthalpy in La_1.9_Ca_0.1_Mo_2_O_8.95_ and La_1.85_Ca_0.15_Mo_2_O_8.925_ samples during both cooling stages (Figure 4). Meanwhile, by the further increase in the calcium substitution degree in the La_2–x_Ca_x_Mo_2_O_9–x/2_ system, the decrease in the phase transition enthalpy is already determined by a significant lack of the La_2_Mo_2_O_9_ crystalline phase. This conclusion is confirmed by the constant increase in the concentration of the crystalline phase of calcium molybdate in the final mixture of the obtained ceramics.

## 4. Conclusions

This study showed that the homogeneous substitution of lanthanum by calcium ions takes place up to the compound of initial composition for La_1.95_Ca_0.05_Mo_2_O_8.975_. In this case, the decrease in the phase transition enthalpy is determined by the increase in the concentration of the formation of the mixed compound for the initial composition of La_2–x_Ca_x_Mo_2_O_9–x/2_. Meanwhile, the influence of the monoclinic phase amount on the phase transition enthalpy remained important only in the case of the formation of a pure La_2_Mo_2_O_9_ compound, the amount of which significantly increases with the appearance of the CaMoO_4_ impurity phase in the ceramic mixture. In summary, it can be concluded that the formation of the impurity of the calcium molybdate crystal phase, which compensates for the lack of lanthanum and the excess of molybdenum in the multicomponent oxide La_2–x_Ca_x_Mo_2_O_9–x/2_ system, has a significant influence on the decrease in the phase transition enthalpy in the La_2_Mo_2_O_9_ compound. The influence of the monoclinic phase amount on the phase transition enthalpy remains an important factor only in the case of the pure lanthanum molybdate.

## Figures and Tables

**Figure 1 materials-16-00813-f001:**
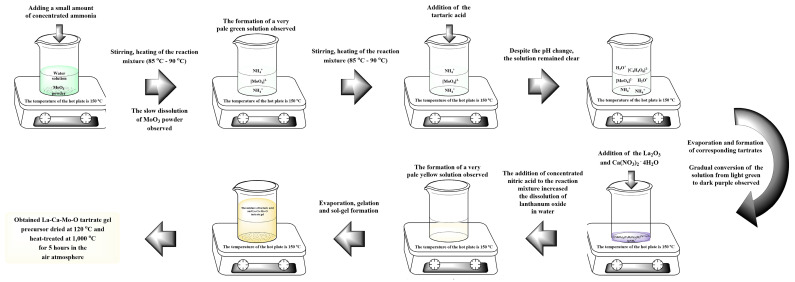
Synthesis scheme of the La–Ca–Mo–O tartrate precursor for La_2–x_Ca_x_Mo_2_O_9–x/2_ ceramic.

**Figure 2 materials-16-00813-f002:**
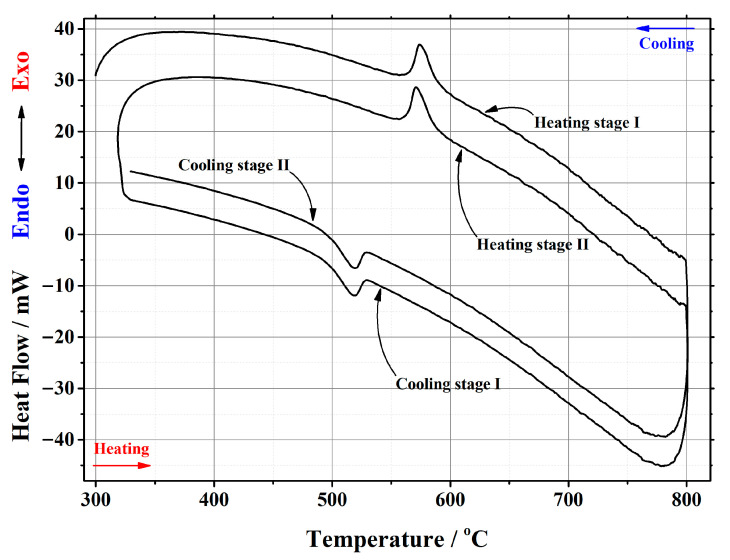
DSC curve of the phase transition cycles for La_1.95_Ca_0.05_Mo_2_O_8.975_ ceramic heat-treated at 1000 °C.

**Figure 3 materials-16-00813-f003:**
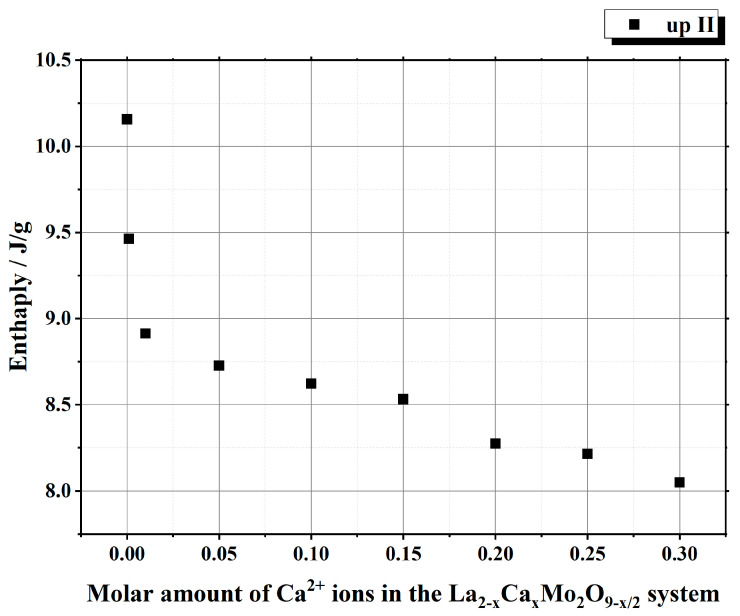
Dependency of the phase transition enthalpy values from the substitution degree by calcium in the La_2–x_Ca_x_Mo_2_O_9–x/2_ system under the second heating stage.

**Figure 4 materials-16-00813-f004:**
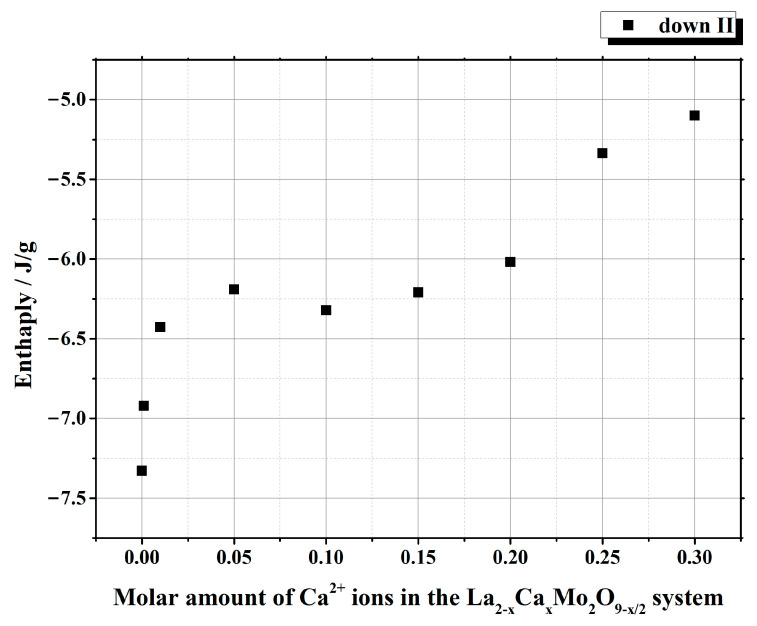
Dependency of the phase transition enthalpy values from the substitution degree by calcium in the La_2–x_Ca_x_Mo_2_O_9–x/2_ system under the second cooling stage.

**Figure 5 materials-16-00813-f005:**
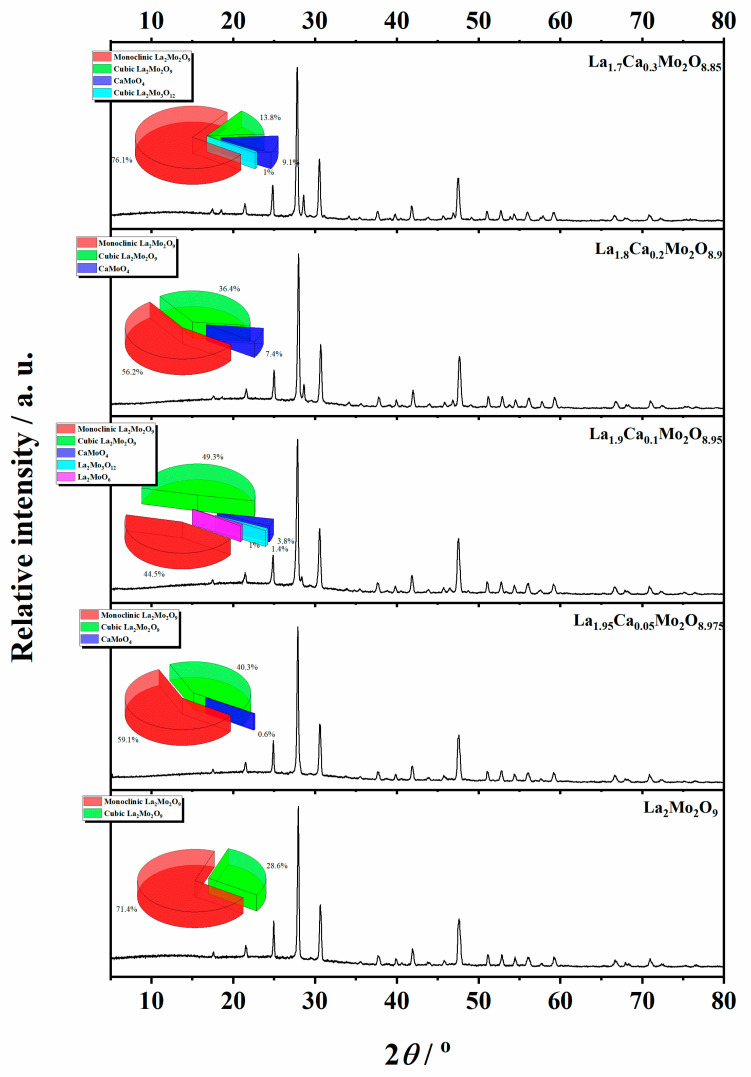
XRD patterns of the La_1–x_Ca_x_Mo_2_O_9–x/2_ ceramic heat-treated at a 1000 °C temperature.

**Table 1 materials-16-00813-t001:** Thermoanalytical data and α↔β phase transition peak properties for La_2–x_Ca_x_Mo_2_O_9–x/2_ ceramic.

Initial Composition	Sample Mass/mg	Heating/Cooling Stages		Temperature/°C			Heat	
				Onset	End	Peak Position	Flow/mJ	Enthalpy/J·g^–1^
La_2_Mo_2_O_9_	20.181	heating	stage I	556.07	579.33	563.25	195.047	9.665
			stage II	555.69	578.84	563.03	204.953	10.1559
		cooling	stage I	540.31	523.80	533.34	–146.255	–7.2473
			stage II	540.26	523.17	533.29	–147.886	–7.3281
La_1.999_Ca_0.001_Mo_2_O_8.9995_	22.158	heating	stage I	558.63	580.56	567.11	191.073	8.6231
			stage II	556.36	578.06	564.68	209.688	9.4632
		cooling	stage I	537.43	511.92	525.65	–156.176	–7.0482
			stage II	537.45	511.92	526.09	–153.356	–6.9209
La_1.99_Ca_0.01_Mo_2_O_8.995_	22.153	heating	stage I	562.12	585.08	569.41	194.41	8.7757
			stage II	559.49	582.62	566.63	197.462	8.9134
		cooling	stage I	533.02	512.79	523.19	–145.764	–6.5798
			stage II	532.63	511.21	522.73	–142.37	–6.4266
La_1.95_Ca_0.05_Mo_2_O_8.975_	22.146	heating	stage I	565.55	587.31	573.98	183.43	8.2826
			stage II	563.28	587.47	571.59	193.262	8.7266
		cooling	stage I	527.46	497.28	517.3	–134.667	–6.0808
			stage II	527.68	498.61	518.05	–137.095	–6.1904
La_1.9_Ca_0.1_Mo_2_O_8.95_	22.183	heating	stage I	567.18	589.94	575.53	182.851	8.2429
			stage II	564.64	587.71	573.60	191.273	8.6225
		cooling	stage I	530.34	512.47	525.64	–139.347	–6.2817
			stage II	530.42	513.60	525.82	–140.215	–6.3208
La_1.85_Ca_0.15_Mo_2_O_8.925_	22.189	heating	stage I	566.04	589.64	574.53	181.146	8.146
			stage II	563.04	588.21	572.27	189.325	8.5323
		cooling	stage I	530.33	515.25	525.78	–137.468	–6.1953
			stage II	530.38	515.97	526.06	–137.764	–6.2086
La_1.8_Ca_0.2_Mo_2_O_8.9_	22.200	heating	stage I	566.1	589.06	574.45	175.753	7.9168
			stage II	563.01	585.8	571.18	183.708	8.2751
		cooling	stage I	532.05	515.17	526.57	–137.915	–6.2124
			stage II	532.12	516.21	526.66	–133.617	–6.0188
La_1.75_Ca_0.25_Mo_2_O_8.875_	22.182	heating	stage I	561.67	586.44	571.56	181.96	8.203
			stage II	547.82	573.6	556.83	182.224	8.2148
		cooling	stage I	526.46	508.12	521.81	–115.632	–5.2128
			stage II	526.48	510.01	521.98	–118.351	–5.3354
La_1.7_Ca_0.3_Mo_2_O_8.85_	22.192	heating	stage I	560.67	585.41	569.80	176.441	7.9507
			stage II	546.08	571.05	555.68	178.614	8.0486
		cooling	stage I	526.23	508.93	521.35	–111.515	–5.0250
			stage II	526.17	510.26	521.13	–113.167	–5.0995

**Table 2 materials-16-00813-t002:** Crystal system, mass fraction, crystallite size, lattice parameters, and agreement indices for the La_2–x_Ca_x_Mo_2_O_9–x/2_ ceramic.

Initial Composition	Crystal Phase	Crystal System	Mass Fraction/%	Crystallite size/nm	Unit Cell			Weighted R Profile	Goodness of Fit
					a/pm	b/pm	c/pm		
					alpha/^o^	beta/^o^	gamma/^o^		
La_2_Mo_2_O_9_	La_2_Mo_2_O_9_	monoclinic	71.4	104.75	1431.438	2145.289	2855.431	12.99106	1.29603
					90.00000	90.42323	90.00000		
	La_2_Mo_2_O_9_	cubic	28.6	47.03	715.106	715.106	715.106		
					90.00000	90.00000	90.00000		
La_1.999_Ca_0.001_Mo_2_O_8.9995_	La_2_Mo_2_O_9_	monoclinic	48.9	66.33	1432.093	2145.928	2857.133	10.70047	1.87511
					90.00000	90.35913	90.00000		
	La_2_Mo_2_O_9_	cubic	50.4	45.56	715.357	715.357	715.357		
					90.00000	90.00000	90.00000		
	CaMoO_4_	tetragonal	0.7	–	–	–	–		
La_1.99_Ca_0.01_Mo_2_O_8.995_	La_2_Mo_2_O_9_	monoclinic	54.1	71.50	1431.437	2145.437	2856.032	10.55389	1.79591
					90.00000	90.38470	90.00000		
	La_2_Mo_2_O_9_	cubic	44.2	46.61	715.103	715.103	715.103		
					90.00000	90.00000	90.00000		
	La_2_Mo_3_O_12_	monoclinic	1.2	41.08	1739.278	1186.510	1624.259		
					90.00000	107.93130	90.00000		
	CaMoO_4_	tetragonal	0.5	–	–	–	–		
La_1.95_Ca_0.05_Mo_2_O_8.975_	La_2_Mo_2_O_9_	monoclinic	59.1	70.52	1431.201	2145.733	2857.156	10.32976	1.76384
					90.00000	90.35389	90.00000		
	La_2_Mo_2_O_9_	cubic	40.3	48.10	715.171	715.171	715.171		
					90.00000	90.00000	90.00000		
	CaMoO_4_	tetragonal	0.6	–	–	–	–		
La_1.9_Ca_0.1_Mo_2_O_8.95_	La_2_Mo_2_O_9_	monoclinic	44.5	35.83	1432.385	2140.825	2855.251	12.83825	2.41047
					90.00000	90.15601	90.00000		
	La_2_Mo_2_O_9_	cubic	49.3	42.06	714.384	714.384	714.384		
					90.00000	90.00000	90.00000		
	CaMoO_4_	tetragonal	3.8	–	–	–	–		
	La_2_Mo_3_O_12_	monoclinic	1.4	43.47	1719.584	1166.525	1614.533		
					90.00000	108.09910	90.00000		
	La_2_MoO_6_	tetragonal	1.0	42.52	582.792	582.792	3031.347		
					90.00000	90.00000	90.00000		
La_1.85_Ca_0.15_Mo_2_O_8.925_	La_2_Mo_2_O_9_	monoclinic	76.0	66.77	1430.812	2144.216	2854.451	16.89944	2.2630
					90.00000	90.36139	90.00000		
	La_2_Mo_2_O_9_	cubic	17.1	44.52	714.631	714.631	714.631		
					90.00000	90.00000	90.00000		
	CaMoO_4_	tetragonal	5.8	59.84	526.101	526.101	1153.607		
					90.00000	90.00000	90.00000		
	La_2_Mo_3_O_12_	monoclinic	1.1	42.68	1732.883	1168.940	1619.405		
					90.00000	107.77000	90.00000		
La_1.8_Ca_0.2_Mo_2_O_8.9_	La_2_Mo_2_O_9_	monoclinic	56.2	45.44	1428.985	2143.602	2858.397	12.46852	2.36196
					90.00000	90.31453	90.00000		
	La_2_Mo_2_O_9_	cubic	36.4	46.98	714.584	714.584	714.584		
					90.00000	90.00000	90.00000		
	CaMoO_4_	tetragonal	7.4	58.26	525.675	525.675	1151.621		
					90.00000	90.00000	90.00000		
La_1.75_Ca_0.25_Mo_2_O_8.875_	La_2_Mo_2_O_9_	monoclinic	79.6	45.63	1430.900	2142.097	2850.290	14.09104	1.54335
					90.00000	90.29116	90.00000		
	La_2_Mo_2_O_9_	cubic	12.2	39.32	714.035	714.035	714.035		
					90.00000	90.00000	90.00000		
	CaMoO_4_	tetragonal	6.4	48.20	523.288	523.288	1146.182		
					90.00000	90.00000	90.00000		
	La_2_Mo_3_O_12_	monoclinic	1.8	46.96	1732.404	1167.824	1617.912		
					90.00000	107.70840	90.00000		
La_1.7_Ca_0.3_Mo_2_O_8.85_	La_2_Mo_2_O_9_	monoclinic	76.1	64.96	1430.166	2143.528	2854.548	13.81435	1.44730
					90.00000	90.34066	90.00000		
	La_2_Mo_2_O_9_	cubic	13.8	45.66	714.447	714.447	714.447		
					90.00000	90.00000	90.00000		
	CaMoO_4_	tetragonal	9.1	67.30	523.476	523.476	1146.807		
					90.00000	90.00000	90.00000		
	La_2_Mo_3_O_12_	monoclinic	1.0	66.99	1733.132	1169.219	1619.159		
					90.00000	107.79630	90.00000		

## Data Availability

The data presented in this study are available on request from the corresponding author.

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
