# Peer review of "Thermoanalytical and X-ray Diffraction Studies on the Phase Transition of the Calcium-Substituted La2Mo2O9 System"

_materials, 2023, doi:10.3390/ma16020813_

Round 1
Reviewer 1 Report
In this manuscript entitled "Thermoanalytical and X-ray diffraction studies on the phase transition of the calcium-substituted La2Mo2O9 system", the authors have studied the dependency of the enthalpy value of phase transition of La2–xCaxMo2O9–x/2 ceramic prepared by an aqueous sol-gel preparation technique via using the thermal analysis combined with the X-Ray diffraction and found that the heat intensity of the phase transition was significantly affected by such factors as the amount of the La2Mo2O9 compound and the concentration of the monoclinic crystalline phase in the final mixture. This paper has some shortcomings, which should be improved.
In the Introduction, the molar ratio of initial metals remains the main factor that determines the formation of the La2Mo2O9 composition. This is the reason why the partial substitution of either lanthanum or molybdenum leads to the crystallization of side phases, which significantly affects the physical properties of the corresponding compound. The related phenomenon can be found in lots of related reports [such as H. Zhang, Y. Wang, H. Wang, D. Huo, W. Tan, J. Appl. Phys. 131 (2022) 043901]. The reference is related to structure, ions doping and phase transition of perovskite oxides, which is suggested to be cited in this paper.
Miller indices of XRD peaks (Figure 5) should be marked in the XRD.
Crystallite size has been listed in Table 1. The detailed method for obtaining grain size is suggested to be described in the paper.
How to obtain the oxygen content of the sample, such as La1.9Ca0.1Mo2O8.95 ? La1.95Ca0.0 5Mo2O8.975 ?
Can the authors provide the SEM measurements to further determine the grain size?
Author Response
First of all, I would like to thank the reviewers for their valuable comments and suggestions that make this manuscript better.
Thank you for the advice on how to increase the number of cited articles in the manuscript. It is a pity that there are not many Chinese scientists who work on the topic of LAMOX compounds.
At least two phase mixtures were identified in all XRD diffractograms, which makes manual indexing quite complicated and imprecise. On the other hand, Rietveld refinement is a much more accurate technique for crystal phase identification than manual Miller index labelling.
X'Pert HighScore Plus software was used to analyze XRD diffractograms and estimate lattice parameters and crystallite sizes.
The amount of oxygen in the compound was determined by estimating the number of cations in the final multicomponent oxide.
The images of the surface morphology do not agree with the results of the XRD analysis due to the phase transformation above the temperature of 500 oC. In this case, the resulting particles are larger than the crystallites determined by XRD analysis. On the other hand, the surface morphology of the La2-xCaxMo2O9-2/x samples with a smaller substitution degree (x=0.001, 0.01, 0.05) is identical, so we did not want to spoil the main idea of the article and overload the manuscript with redundant information.
Reviewer 2 Report
Dear Authors! Thank you for your manuscript, submitted in "Materials". After the reading of your article, devoted to the DSC and XRD study of the Ca-doped La2Mo2O9 complex oxides, I have the following comments:
1. Abstract is so voluminous and blank. It needs in the strong shortening. Besides, the fact data would be noted, and the main conclusion of the work would be highligthed.
2. It is not clear from the Introduction Part, why was Ca chosen as a dopant?
3. Introduction does not fullly overview the present literature picture, connected with the thermal srability and the temperature range of the La2Mo2O9 complex oxides with various structure.
4. The obtained results needs strongly in Discussion. There are absolutely no the references in the Part of Results and Discussion: The manuscript contains 28 references, among them - 26 in Introduction, 2 in Experimental Part.
5. The compositions of the obtained samples would be noticed in the Experimental Part.
6. It seems the strange the obtaining of the sample with x=0.001 - weighing is less than the accuracy of an analytical balance.
7. Line 109. In the top paragraphes there is no information about the amount of monoclinic phase (and other phases) in the sample. Please, clearify this fact.
8. Line 109. x=0.5. There is typo?
9. The existence of the CaMoO4 phase in all doped samples was observed. It seems to be the absence of Ca-intercalation into La2Mo2O9 oxide structure. In opposite, it needs in approvement, for example, with the calculations of the cell parameters of undoped and doped samples.
10. XRD study is unformative in this case - metastable phases, existing at high temperatues, are not investigated using the above-mentioned method. Data of high-temperature X-Ray diffraction study in-situ is needed for the analysis the phase compositions.
Author Response
First of all, I would like to thank the reviewers for their valuable comments and suggestions that make this manuscript better.
I agree with the opinion of the reviewer that less is better than more. We tried to present less information in this manuscript, but it seems that some aspects are too comprehensive.
The main aim of the manuscript is related to the reduction of the phase transition in the LAMOX system. The origin of the element substituting the lanthanum is not the most important factor in this case.
We did not want to deviate from the essence of the main aim of this manuscript. As you mentioned, less is better than more.
There are no scientists who have synthesized LAMOX compounds by this method, so the discussion of this chapter based on intensive citation is not extensive.
There is no point in repeating the same information many times in all sections.
With great respect, it seems that the reviewer has not often encountered the synthesis of chemical compounds. The value of x=0.001 (La2-xCaxMo2O9-2/x) corresponds to the 0,23615 g of Ca(NO3)2*4H2O. If we take for example 0,0065 mol of the final compound the amount of calcium nitrate tetrahydrate remains significant for scales, which weighs 0.0001 g in accuracy and is equal to 0,0015 g.
We have not been able to determine the substance of this question. Could the reviewer clarify and expand his question?
We agree on existing typing errors. The corresponding changes were made in the text.
We think that a small amount of calcium ions in the corresponding system only slightly affects the molar ratio of monoclinic and cubic phases. This effect is seen from the negligible change in the lattice parameters in the monoclinic system. The formation of the other crystalline phases at high temperatures is less likely due to the low crystallinity of the main cubic phase and the relatively small amount of impurities. The low crystallinity of the cubic La2Mo2O9 system at high temperatures was shown in our previous publication: Kežionis, A., Petrulionis, D., Kazakevičius, E., Kazlauskas, S., Žalga, A., & Juškėnas, R. (2016). Charge carrier relaxation phenomena and phase transition in La2Mo2O9 ceramics investigated by broadband impedance spectroscopy. Electrochimica Acta, 213, 306-313.
An excess of molybdenum leads to the formation of CaMoO4, which proved for the samples with increased substitution. In our previous publications, we also demonstrated that the crystallization of CaMoO4 starts from the temperature of 350 oC, while the crystallization of La2Mo2O9 begins above 500 oC. Thus, the formation of a mixed oxide for La2-xCaxMo2O9-2/x composition is unlikely.
Round 2
Reviewer 1 Report
The paper can be accepted.
Author Response
Dear Reviewer, Thank you for your valuable comments and positive decision for our manuscript's possible publication in Materials.
Reviewer 2 Report
Dear Authors! Thank you for your attention to my comments. Typos in manuscript were corrected. However, I consider, that obtained results are needed in the additional discussion. In opposite, there is not the scientific Article, it looks like the scientific Report.
I would like to propose to include into the consideration the recent (2022-2022) relevant publications, devoted to the properties of La2Mo2O9 - it will allow to highligth the actuality of topic and to extend the discussion, aimed in increasing of readers interest.
DOI:10.1002/ejic.202200165
DOI:10.1021/acsami.1c20839
DOI:10.1680/jnaen.21.00010
DOI:10.1080/00150193.2022.2034444
DOI:10.3390/ceramics4030037
DOI:10.1016/j.ssi.2020.115405
Author Response
Dear reviewer, thank you for your valuable comments. Your suggested articles have been added to the manuscript's reference list.